# Impact of Particular Stages of the Manufacturing Process on the Reliability of Flexible Printed Circuits

**DOI:** 10.3390/s25010140

**Published:** 2024-12-29

**Authors:** Andrzej Kiernich, Jerzy Kalenik, Wojciech Stęplewski, Marek Kościelski, Aneta Chołaj

**Affiliations:** 1Łukasiewicz Research Network—Tele and Radio Research Institute, Ratuszowa 11, 03-450 Warsaw, Poland; wojciech.steplewski@itr.lukasiewicz.gov.pl (W.S.); marek.koscielski@itr.lukasiewicz.gov.pl (M.K.); aneta.cholaj@itr.lukasiewicz.gov.pl (A.C.); 2Faculty of Electronics and Information Technology, Warsaw University of Technology, Nowowiejska 15/19, 00-665 Warsaw, Poland; jerzy.kalenik@pw.edu.pl

**Keywords:** environmental tests, Interconnect Stress Test, design of experiment, Taguchi method, flexible printed circuit, product reliability, production process

## Abstract

The purpose of the experiment was to indicate which element of the production process of flexible printed circuit boards is optimal in terms of the reliability of final products. According to the Taguchi method, in the experiment, five factors with two levels each were chosen for the subsequent analysis. These included the number of conductive layers, the thickness of the laminate layer, the type of the laminate, the diameter of the plated holes, and the current density in the galvanic bath. The reliability of the PCBs in the produced variations was verified using the Interconnect Stress Test environmental test. The qualitatively best variant of the board construction was indicated using the signal-to-noise ratio and the analysis of variance method for each factor. The factors determined to be the most important in terms of reliability were the number of conductive layers and the current density in the galvanic bath. The optimal variant of the board construction was two conductive layers on a polyimide laminate, where the laminate layer was 100 μm thick, the hole diameter was equal to 0.4 mm, and current density was 2 A/dm^2^ in the galvanic bath. Therefore, the plated experiment indicated the factors needed to obtain a high-quality product with a low failure rate.

## 1. Introduction

In the 20th century, Japanese engineer Genichi Taguchi said that “quality is the virtue of design” [1]. The principal challenge for engineers is to design and produce devices that meet the expectations and needs of the end users. Nevertheless, a task of equal importance is the production of a high-quality product. Any high-standard system not only functions properly as expected but also functions over a long period of time without any faults. It can be said that a robust product is reliable, useful, functional, easy to maintain, and efficient. With this in mind, it is possible to fully understand what Taguchi meant in the above quote. Furthermore, high-quality and reliable technology is important for further development of science and new innovations. Ambitious space programs and long-term scientific programs must be supported by the tools they use. Once again referring to Taguchi’s philosophy, the product should not only be free of any defects when leaving the production line, but its high quality during its use should be ensured. The reliability of the device must be maintained regardless of operating conditions throughout its whole nominal period of use. In unfavorable circumstances, which can be called noise, the product should still meet the requirements. It is worth adding that designers are responsible for the robustness of the product. Regardless of how technologically advanced the factory is, mediocre designs will always result in mediocre products [1].

Unfortunately, flawed, low-quality products, with some even designed to fail over time, are becoming a bigger issue every day. This problem has two sides: ecological and economical. The issue of poor-quality products concerns all industries; however, herein, we focus on the electronics sector. According to the Sustainable Development Goals 2020 report issued by the United Nations, every year for the last decade, the average resident of our planet has produced from 5.3 to 7.3 kg of e-waste [2]. The amount of waste is constantly growing. Equally alarming is the fact that globally, less than 20% of e-waste is recycled. Raw materials such as copper, silver, gold, and rare-earth metals that are used in electronics are irretrievably lost. Moreover, every year, devices worth billions of dollars are thrown into the trash. This particularly affects the pockets of individual customers, who are forced to buy a new product because the previous one broke down very quickly.

To address the above issue, an experiment was conducted, which constitutes the basis of this paper. The aim of the research was to answer the following question: how do particular stages that occur in the production process affect the aging of electrical connections on flexible printed circuit boards (PCBs)? The connections were fabricated by metallizing holes. The Interconnect Stress Test (IST) was selected and used as an aging test during the experiment. This method made it possible to determine the resistance of the printed circuit board to rigorous assembly techniques or the operating conditions of an electronic device. The experiment was planned according to the Taguchi method, which allowed us to reduce the number of tests and determine the most favorable factors during the production process. The research results enabled the selection of optimized conditions and factors occurring in the process of manufacturing end products involving the Internet of Things and critical applications of Industry 4.0, including sensors.

Flexible printed circuit (FPC) technology has been present in the electronics industry for several decades. It is a response to the miniaturization and growing quality requirements of electronic equipment. Printed circuit boards made of flexible materials, such as polyimide and polyesters, offer design freedom in electronic devices, and save space, thereby reducing the size and weight of electronic equipment. Flexible laminates have another great advantage that distinguishes them from traditional, rigid glass–epoxy laminates. They solve the severe problem of the mismatch of coefficients of thermal expansion (CTE) of laminate and copper in plated through-holes (PTHs). Differences in the degrees of expansion of materials leads to excess stress generated in the components and electronic connections. In the worst case, this causes their breakage and circuit failure. However, the usage of flexible substrate, even if it is not fully plastic due to the assembly of components or the use of a connection with a rigid laminate, can reduce the effect of the mismatch of coefficients of thermal expansion while maintaining high reliability and low costs [3,4]. Flexible printed circuit boards have found application in relatively small devices such as notebooks, smartphones, and, thanks to its elasticity, wearable devices [5,6].

As was previously noted, the reliability of products should be a key requirement of their manufacture. At the prototyping and testing stage, this is checked in many ways. In the case of printed circuit boards, these are not just functionality tests. PCBs are often used in critical applications (in medical, space, military, or automotive applications and high-power devices) and work in demanding conditions (such as high-humidity environments or where mechanical shocks or rapid temperature changes occur). To check how PCBs behave in such conditions, they are subjected to many environmental tests. In order to ensure devices’ safety, reliability, and compliance with the requirements and standards, products should successfully pass the analyses mentioned above. All tests should be performed in accordance with the IEC 60068 standard, which specifies a number of environmental testing methods. It specifies various atmospheric conditions for measurement to assess the ability of samples to perform under diverse scenarios [7]. These tests include accelerated HALT/HASS tests, water absorption tests (in a highly humid environment), vibration tests, and thermal shock tests. Each of them allows us to verify the device from a different perspective for particular assignments.

In this research, the Interconnect Stress Tests were selected as a methodology to check the reliability of flexible printed circuits. The IST, also known as thermal cycling induced by direct current testing, is an accelerated aging test method developed in the 1990s to measure changes in the resistance of plated through-holes and reliability of interconnection in printed circuit boards. They arose from the need for a fast, repeatable and reproducible inspection, the results of which would additionally correspond to those obtained from the existing methods. Moreover, the new technology was supposed to make it cheap and easy to analyze the results. The idea of the IST is based on the flow of direct current through the plated through-holes on the board. The current heats the metallization and the surrounding laminate. Eventually, as a result of the cyclic heating and cooling of the PCB, a failure occurs. The primary cause of the failure in ISTs is considered to be a mismatch in the coefficient of thermal expansion between the epoxy resin and the fiberglass in the substrate materials, and the copper in the through-holes. The effect of the failure is a crack in the metallization of plated holes. The IST relies on specially designed sample coupons placed on PCB fabrication panels [8,9,10,11].

Environmental tests, especially the IST, allow us to observe the behavior of individual technological solutions in high-power applications, such as GaN semiconductor technology [12]. The aim of these activities is, among others, the production of circuits with high electrical and thermal reliability for modern solutions that include GaN technology. This aligns with the global trends in the development of so-called Green Power Electronics. The efficient power supply and charging of electronic devices and electric vehicle batteries is now becoming a necessity. Combining the efficiency of the GaN with the resistance of flexible materials allows for significant increases in productivity and reliability.

This particular study investigated the reliability of flexible printed circuits by environmental thermal tests. This research included samples differing in their number of conductive layers, the type and thickness of the laminate used, the diameter of the metallized holes, and those made using different values of current density in the electroplating copper plating bath. The use of various production factors was aimed at determining which of them had the greatest impact on the reliability of the tested samples. The qualitatively best variant of the board construction was indicated using a signal-to-noise ratio (SNR) and an analysis of variance (ANOVA) method for each factor. To better understand the processes present in the tested PCBs, samples were analyzed by a metallographic cross-section, along with an observation under a scanning microscope and an infrared camera.

## 2. Materials and Methods

### 2.1. Selection of Experimental Factors

The production process of printed circuit boards is extremely complex. It consists of many stages and is influenced by many different factors. The path through which the PCB passes in the factory from the raw laminate to the finished product is shown in Figure 1. The first step in designing the experiment is to select several factors among all others. This is necessary due to the multitude of elements in the PCB production process and the limited time and material resources. In this paper, the focus was on factors that can meet the requirement of flexibility in the board and of elements of the PCB structure playing their critical roles during the Interconnect Stress Test. During the selection of the elements that would be a part of the experiment, decisions that were directly influenced by the designer of PCBs (or after a consultation with the producer) were taken into account.

Five factors that may have gad the biggest impact on the life of flexible printed circuits were chosen. These were as follows:Type of the laminate: polyimide and glass–epoxy (FR4) laminate;Number of conductive layers: 2 and 4 layers;Thickness of the laminate: 50 and 100 µm;Diameter of the plated holes: 0.3 and 0.4 mm;Current density in the galvanic bath: 1 and 2 A/dm^2^—constant thickness of metallization equal to 20 µm.

#### 2.1.1. Type of the Laminate

The first choice for the base material for FPCs is polyimide, a polymer plastic material, as it is one of the basic laminates used in the production of flexible printed circuits. This is due to the flexibility of the material and its durability when working at higher temperatures. However, a great disadvantage of this material is its high price [13]. In order to compare its performance in the thermal stress tests, we decided to analyze samples produced on glass–epoxy laminate in its most popular variant, FR4 [13]. The problem with glass–epoxy laminate is its rigidity. This is due to subsequent layers of glass fibers being stacked on one another. However, if enough thin base material is used, then the laminate should be flexible and meet the requirements of the application. Bearing this in mind, the purpose of comparing these two base materials was to investigate whether a traditionally used laminate would endure environmental conditions comparable to those of polyimide material.

#### 2.1.2. Number of Conductive Layers

The number of conductive layers, which is two, among which copper plays the role of the conductor, is dictated by the nature of IST analysis. It is possible to design and fabricate plated through-holes only when the PCB has no less than two layers, where the conductor is located on both sides of the laminate. Currently, in advanced systems where integrated circuits with very numerous pinouts or a ball grid array (BGA) package are used, it is necessary to design multi-layer boards. Despite the increased complexity of the structure and the production process, this solution has many advantages [14]. The second simplest PCB design after the double-layer board is the four-layer board. Therefore, the reliability of the device was checked depending on the complexity of construction of a PCB.

#### 2.1.3. Thickness of the Laminate

The most flexible films come in a narrow range of material thickness from 12 μm to 125 μm [3]. Hence, we decided to arbitrarily chose two values from the middle of the range, 50 μm and 100 μm. It is worth noticing that the practice shows that thinner laminates cause more problems during PCB production. It is necessary to achieve a reasonable tradeoff between the application requirements and the manufacturer’s capabilities. The selection of the material’s thickness as the next factor allowed us to check whether thin laminate is more resistant to thermal exposure. The thickness of a single laminate layer and the number of conductive layers together affect the total thickness of the PCB.

#### 2.1.4. Diameter of the Plated Hole

The diameter of the plated holes varies a lot, from 100 µm to several millimeters. This is due to the many roles that PTHs play. They can function as an electronic interconnections between different parts of the circuit, heat sinks, and mechanical supports for component assembly, among many other purposes. As the size of the plated through-holes decreases, their durability also decreases. On the other hand, the reliability of the vias is highly influenced by the ratio of its diameter to its depth (the smaller the ratio, the more durable the connection). [15] It was also important to select commonly used hole sizes [16]. Bearing all this in mind, to maximize the potential of thermal stress testing, we decided to create holes measuring 0.3 mm and 0.4 mm.

#### 2.1.5. Current Density in the Galvanic Bath

Shortening the production time of a single product is the top priority of every manufacturer. In the case of PCB production, one way to achieve this goal is by controlling the current density value during the electroplating process. The higher the current density, the faster the process of copper deposition [17]. Increasing the current density to the maximum possible value is a common practice in production plants. Thanks to this, the bath time can be made shorter, and a larger number of printed circuits can be produced. However, alternating the current density changes the crystal structure of the deposited metal. Because of this, the last element of the production process that was been taken into account when planning this experiment was the metallization of holes in the galvanic bath. This is due to the electrochemical deposition of a metal coating into a hole drilled into a laminate using a direct electric current. The coated surface serves as the cathode of the electrolytic cell. The anode is a block of a conductive material [13]. Copper is deposited by cation reduction. After consultation with the factory that produced the test samples, we decided to choose two metallization variants using a current density of 1 A/dm^2^ and 2 A/dm^2^. The aim was to find an answer to the question of whether the shape of copper crystals in the holes has an impact on reliability.

The selected factors and their levels are shown in Table 1 below.

### 2.2. Design of the Experiment

The experiment was designed and planned according to the method proposed by Genichi Taguchi. Designing experiments is a part of his entire approach to addressing the problem of product quality loss. The basic concept of Taguchi’s philosophy is achieving a reduction in system instability at its output and therefore an improvement in the device’s quality. When the observed characteristics at the output of the received product differ from the ideal or the desired ones, quality loss is considered to have occurred. With regards to quality, Taguchi introduces the classic signal-to-noise ratio (SNR) [18,19]. The signal power below is the desired response, and the noise power is the deviation from the ideal product characteristics:(1)$$\mathrm{SNR}=10\log_{10}\frac{\mathrm{power}\;\mathrm{of}\;\mathrm{signal}}{\mathrm{power}\;\mathrm{of}\;\mathrm{noise}},$$

The best design is not one that maximizes the signal strength, but rather one that effectively reduces the impact of the noise. The SNR value should be as high as possible regardless of external factors that uncontrollably affect the system. Therefore, the goal of quality improvement efforts is to achieve a maximum signal-to-noise ratio. The aim of Taguchi’s method is to reduce the variability in the final product under different conditions in the work environment. This ensures that the choices made in the laboratory also remain valid during production and use [18,19].

Considering the theory of quality loss, Taguchi proposes a way to plan the experiment. His idea is based on the partial–factorial design of experiments (DOEs) and the planning of subsequent studies based on an orthogonal array [18,19]. Thanks to this, Taguchi’s method reduces the number of test runs required to perform. The orthogonal arrays are an extension of Latin squares. Written in the following form, L_N_(s^m^), they are matrices of n rows by m columns. A specific property of orthogonal arrays is that in each pair of columns, each of the possible ordered pairs of elements appears the same number of times. The columns correspond to individual factors, the contents of cells in the columns correspond to factor levels and the rows correspond to test runs. Taguchi suggested a catalog of 20 arrays (including 18 orthogonal ones), but over time, the concept was developed and further suggestions were given. It is also possible to create submatrices by removing some columns from the array. In this way, many different multivariate experimental designs can be generated [20].

The experiment was conducted based on the selected criteria. For a full-factorial design of the experiment, 2^5^ = 32 tests would have been necessary to perform. We decided to design the experiment by using fractional-factorial design, and the Taguchi method was chosen for this purpose. This way, the number of tests required to perform the research was reduced. The first step in designing the experiment according to Taguchi is to construct an orthogonal array. As was explained in the previous paragraph, because 5 factors with 2 levels each were selected, we decided to use an L_8_(2^7^) orthogonal array (array L_8_ had 8 rows and 7 columns, so in case of the research described in this paper, 2 columns remained unused). Therefore, the entire experiment was carried out on the basis of 8 tests. The orthogonal array constructed for the purpose of the experiment is shown in Table 2. The 8 test runs, allowed for time savings, and this was achieved via the use of the fractional-factorial design for experiments according to the Taguchi’s method, instead of the full-factorial design of experiments. This led to a huge reduction in costs in terms of time and material intake. In order to eliminate accidental errors that could have distorted the statistics obtained at the end of the study, from each produced PCB construction variant, three copies were drawn and tested within the experiment.

### 2.3. Design of Printed Circuit Board

The tests performed for the purpose of this experiment required specifically designed and manufactured PCBs. Samples were not only fabricated in accordance with the selected factors but also had to be adequate for the IST. The IPC (formerly the Institute for Printed Circuits) standard number 650-TM 2.6.26 provides guidelines on how the samples should be designed. [21] The design of samples is shown in Figure 2, and an example of a PCB used as the sample is shown in Figure 3.

The standard distinguishes between two possibilities of a track routing. In the first method, named “Method A”, there are two or more independent tracks on the test board. The power supply and heating circuit is marked as POWER, and the measuring circuit’s resistance is marked as SENSE. In both methods, direct current flowing through the tracks heats the entire board to the set temperature, which then is cooled to room temperature. However, “Method A” was chosen because it allows for the measurement of the resistance in both the sensing and power supply circuits [21].

The basic design requirement for the samples was the dense arrangement of a large number of connections between the board layers. The aim of such a solution was to maximize the chance of a failure. The distance between successive metallized holes was 1.25 mm. The holes were planned along the entire length of the board. The power supply and measurement circuits were arranged to resemble a braid.

When designing the board, a decision was made to use a laminate with a 17.5 µm thick copper foil layer. In the technological process of producing the printed circuit mosaic, another copper layer measuring approximately 17.5 µm was applied to the tracks. This made the total thickness of a single track approximately 35 μm. The thickness of the metallization in the holes was about 20 µm as a result of electroplating. As was previously mentioned, one of the characteristic parameters of the electroplating process of copper is its duration. In the production of the test boards, previously selected current densities were used in two variants. At a current density of 1 A/dm^2^ and with an efficiency of 100%, 0.2 μm of copper was deposited in one minute. For the established galvanic bath parameters, the deposition of the desired copper thickness took 110 min for 1 A/dm^2^ and 55 min for 2 A/dm^2^. As can be observed, increasing the current density significantly affects the production time.

Each PCB measured 14 mm × 120 mm. The standard recommends that the test sample has the shape of a rectangle, usually with dimensions of 0.6 and 5 inches, that is, respectively, 1.524 and 12.7 cm. They should have pinouts for the power supply and measurement circuits on both short edges, as well as male connectors with a spacing of 2.54 mm. [21] The thickness of the PCBs of different variants along with the rest of the information about test samples is presented in Table 3. In the case of the laminate made of polyimide, a material from the DuPont manufacturer (Wilmington, DE, USA) Pyralux AP series was used. Depending on the desired laminate thickness, AP8525R or AP8545R laminate was used (50 μm or 100 μm, respectively).

#### Microvias

During the design of the test boards, the concept of used types of connections was clarified. In the case of a two-layer board, plated through-holes were used, while in four-layer boards, blind microvias were used. This design was chosen because it enabled us to test these two solutions and check them for durability and robustness.

A micro-hole, or microvia, is a blind one, and according to the definition of the IPC-T-50 standard, is a structure with a maximum aspect ratio of 1:1; it can have a depth of no more than 0.25 mm [22]. Blind micro-holes are characterized by a thinner layer of metallization near the neck of the microvia. This is the place where cracks most often occur. This happens because the expanding substrate presses down onto the neck of the microvia. For this reason, producing PCBs with this solution gave us an idea of the difference between the robustness of microvias and of PTHs.

### 2.4. Performed Aging Tests

The entire IST procedure with details of individual stages and explanations are provided below.

The first step in the sample testing procedure was to measure the resistance of the POWER and SENSE circuits. Then, the exposure current was determined. The resistance values of the circuits on the boards differed slightly, which necessitated individual determinations of the exposure current for each sample. According to the IPC-TM-650 standard, 30-s tests, 60-s tests and pre-tests could be performed in order to adjust the exposure current. The 30-s and 60-s tests were used to pre-adjust the exposure currents to prevent the sample from heating up too quickly. Any number of 30-s and 60-s tests could be performed. After the initial estimation of the exposure currents, it was necessary to proceed to pre-tests. The sample was under full exposure for 3 min. If the sample reached the required temperature, the process of determining the exposure current was considered completed and the sample was accepted for subsequent cycles. The temperature reached in 180 s should be ±1 °C of the nominal value specified in the IPC standard [21].

The full exposure of samples could be progressively achieved after receiving a positive result from the pre-test. The maximum number of cycles was set to 10,000, and the possible relative change in resistance at the maximum and ambient temperature was set to 10%. The heating time in a single cycle was 180 s. After 3 min, the sample was cooled to room temperature by forced air. The cooling time was a function of the overall thickness and design of the board. Following the suggestion from the standard IPC-650-TM from 1999, the maximum cooling time was set to 2 min [11]. The resistance was measured every 2 s during heating in both circuits, POWER and SENSE; however, during cooling, it was only measured in the SENSE circuit.

When the change in the resistance crossed the threshold of 10% or the 10,000^th^ cycle was reached, the process of sample exposure stopped. The measurement system recognized six different possible reasons for the test’s termination. The first one was the successful completion of all planned exposure cycles. The remaining five determined reasons for interrupting the test were as follows: a break in the POWER circuit, a break in the SENSE circuit, the maximum temperature being exceeded, the dR (acceptable resistance change) at ambient temperature being exceeded, and dR being exceeded at the maximum temperature. Throughout the cycle, the system compared the current resistance with the value measured during the first cycle at the maximum and ambient temperatures. If the change in resistance during heating was more than ten times the initial value, the system considered it a break in the POWER or SENSE circuit. If the change was one and a half times greater than the acceptable value at the maximum temperature, it was treated as a temperature exceedance. However, if at the end of the heating or cooling process, the change in the resistance exceeded the acceptable value, the test was terminated due to the dR being exceeded at the maximum or room temperature.

The sample temperature was calculated based on the measured resistance using the following formula:(2)$$T=\frac{\frac{R_{T}}{R_{\mathrm{env}}}-1}{\alpha }+T_{\mathrm{env}},$$
where
T—temperature of POWER or SENSE circuit;α—temperature coefficient (for copper it is equal to $3.9\times 10^{-3}\;1/K$);R_T_—resistance at temperature T;R_env_—resistance at room temperature;T_env_—ambient temperature.


The summary of the IST procedure is presented in Table 4 below.

#### Workstation

The workstation was composed of the following elements:Control computer—laptop from DELL (Round Rock, TX, USA);Six-channel digital multimeter—3706A-NFP from Keithley (Cleveland, OH, USA);Power supply I—HMP2030 from Rohde and Schwarz (Munich, Germany);Power supply II—HMP2040 from Rohde and Schwarz (Munich, Germany);Temperature sensor—USB-Tset electronic thermometer from Aqua Lab (Warsaw, Poland);12 V computer fan from Arctic (Brunswick, Germany).

The workstation is presented in Figure 4. The control unit was connected to other devices via a USB interface. The resistance in the SENSE circuit was measured with an ohmmeter in the range of 5 Ω with a resolution of 1 mΩ. During heating, the resistance of the POWER circuit was measured using the indirect method by reading the current and voltage (with a resolution of 1 mA and 1 mV, respectively), and then performing a resistance calculation. Eight samples were tested simultaneously. Each test board was cooled individually by its own fan, and each POWER circuit was individually powered from a programmable power supply. In order to minimize the impact of the external environment on the test result, the station was placed in an air-conditioned room, and the sample and the fan were covered with a screen.

## 3. Results

The Interconnect Stress Test results are presented in Figure 5. The test results are grouped into eight clusters that correspond to each of the board construction variants.

Figure 5 shows the results of the experiment in an illustrative way. However, just by observing the graph itself, it can be seen that in the case of the variant of two-layer boards with plated through-holes, the number of cycles achieved was significantly higher than in the case of the construction with microvias. It can also be seen that the smallest deviation in results was achieved in the fifth experiment, while the largest was achieved in the third. Table 5 presents the basic statistical data. The arithmetic mean for each variant, independently and for the entire population, was calculated. The standard deviation for each variant was also provided, and the signal-to-noise ratio was calculated according to Taguchi’s concept. In the conducted experiment, the more cycles the test board survived, the better it was; therefore, the formula for SNR is as follows:(3)$$\mathrm{SNR}=10\log_{10}\left(\frac{1}{n}\sum_{i}^{n}\frac{1}{y_{i}^{2}}\right),$$
where

n—number of samples;y_i_—result of a sample number, i.

An integral part of the analysis of the results is the calculation of the main effects, independent factors influencing the dependent variable. In the case of the conducted tests, this was the effect of the individual parameters of the production process on the durability of the test boards. Figure 6a, Figure 7a, Figure 8a, Figure 9a and Figure 10a contain tables of the main effect, their arithmetic mean values and the SNR coefficient for individual cases. The optimal parameter value is highlighted in bold. Figure 6b, Figure 7b, Figure 8b, Figure 9b and Figure 10b show illustrations of the analysis results above. They show the trajectory of the SNR parameter depending on the level of partial factors. 

Based on the obtained results and the above data, it can be concluded that the optimal variant is a two-layer printed circuit board, manufactured on a polyimide laminate, with a laminate layer measuring 100 μm thick, a hole diameter of 0.4 mm and a current density of 2 A/dm^2^ in the galvanic bath. The factors that have an upward trend are the laminate thickness, the diameter of holes and the current density in the galvanic bath. This means that the higher the value of a given parameter, the better the signal-to-noise ratio. In addition, the above graphs show that metallized through-holes and a polyimide laminate perform better than microvias and a glass–epoxy laminate.

In addition to the study proposed by Taguchi, a complex mathematical instrument, the analysis of variance, was performed. This tool allows researchers to assess the significance of differences between the means of individual groups. Conducting this type of analysis allows for a determination of the correlation between selected parameters of the production process and the reliability of flexible printed circuits. Moreover, ANOVA allows for a consideration of cases with many independent factors [23]. In the case of this research, an analysis was carried out for five factors. The calculation results are presented in Table 6.

For a significance level of 99%, the F statistic in all considered cases takes the value of 8.285 [24]. Based on the observation of the data from the table above, it can be concluded that the durability of the connections is equally influenced by two factors: the number of layers (type of connection) and the current density. For a lower significance level, only 90%, the F statistic has a value equal to 3.00698, as read from the table [24]. The final result is significantly influenced by two more factors: the laminate thickness and the hole diameter. The type of laminate does not have a significant effect on the final result.

In the case of the most important factors, the number of conductive layers and the metallization warrant, the ANOVA confirms the results obtained after the analysis using the Taguchi method. Additionally, the percentage share of each factor in the overall scatter of observation values is given in column P. These values once again confirm the previous results.

### 3.1. Additional Research

The aim of additional tests carried out on test boards was to indicate the cause or the effect of the connection failures on the tested printed circuit boards. As part of the tests, photos were taken with a thermal camera, metallographic cross-sections were made and the surface of the metallized holes was observed using a scanning electron microscope.

#### 3.1.1. Analysis by Thermal Camera

The temperature to which the sample heated up was checked using a thermal camera. This device images the recorded infrared radiation emitted by a tested object. This test served as a control. It also allowed us to observe how the dissipated heat was distributed on the test board. Below are thermographic images. Figure 11 and Figure 12 show the sample during heating and the failure location. Thermal imaging camera A320 from Teledyne FLIR (Wilsonville, OR, USA) was used for the tests.

The first conclusion that comes to mind is that the highest temperature is reached in the middle of the board, as can be seen in Figure 11. The dissipated heat was distributed evenly along the entire length of the sample. The measurement performed with the thermal camera also made it possible to make sure that the measurement system was working properly and that the samples were heating up to the planned temperature. This examination also made it possible to locate the failure spot. The resistance increased in the place where the metallization was damaged. The thermographic image shown in Figure 12 clearly shows a hotter spot on the surface of the test board. Even at low exposure currents, with the current at a specific order of magnitude smaller current used during the tests, the difference was sufficient to select a place for a later cross-section to assess the degree of damage.

#### 3.1.2. Analysis by Metallographic Cross-Section

In order to analyze the places of metallization cracks as a result of stresses caused by IST aging tests, metallographic cross-sections of selected test boards were performed. Based on the results of observations with a thermal camera, places were selected where there was a high probability of failure of the metallized connection occurring. Cross-sections were made for the indicated places. The samples for analysis were prepared on the MECAPOL P262 metallographic cross-section unit, Presi (Eybens, France), and the MetaSery 250, BUEHLER (Lake Bluff, IL, USA). Observations were carried out on the Eclipse L-150 metallographic microscope from Nikon (Tokyo, Japan), and the VHX6000 digital microscope from Keyence (Osaka, Japan). The analysis results are presented in Figure 13 and Figure 14. The evaluation of metallographic sections was based on the criteria suggested by the IPC-A-600 standard [25].

All of the cross-sections clearly show the place of the crack and the damage to the metallization in the plated holes. Characteristic failure locations can also be observed on the cross-sections. For a PTH, this is the point of contact between the plated hole and the copper ring on the board surface. For a microvia, this is the so-called neck. The sections also show that in the case of PTHs, the crack starts from the “center” of the board. A similar observation could not be made for a microvia due to the complete damage to the connection.

In addition to the damage to the metallization resulting from the cracks, another effect of thermal stress and of failure can also be observed on the cross-sections. On the damaged boards, laminate delamination occurred, entailing the separation of individual components of the laminate from each other (fibers from resin or Kapton from glue). Voids and cracks in the laminate and the complete separation of individual components of the base material from each other can be observed in Figure 13a. In the case of PTHs, another defect occurred: lifted land above the board surface. A void appeared between the annular ring and the laminate. All the defects described above disqualify the printed circuits from further use.

#### 3.1.3. SEM/EDX

Further tests were carried out using a scanning electron microscope, SEM JSM-7600F, with EDX spectrometry from Jeol (Tokyo, Japan). Cross-sections were made from two samples of PTHs with metallization obtained after bathing with a current density of 1 A/dm^2^ and 2 A/dm^2^, and then copper was etched using a solution of ammonium hydroxide with hydrogen peroxide. Cross-sections were made using a metallographic sample preparation system, type MECAPOL P262, PRESI (Eybens, France), and MetaServ 250, BUEHLER (Lake Bluff, IL, USA). Thanks to this, it was possible to observe the structure of individual copper layers on the laminate. Figure 15 shows layers of copper foil and electroplated with 1 A/dm^2^. Figure 16 shows copper layers with the second current density variant. The tests were performed on samples that were not subjected to the thermal stresses.

In the case of both samples, it is easy to distinguish between the copper that covered the plate in the galvanic bath and the copper foil that the manufacturer applied to the laminate. The latter has large crystals, with the same shape and structure on both samples. In the case of copper applied galvanically, the metal crystals are much smaller, creating a fine structure. The crystals of copper applied galvanically are several times smaller than those from copper foil. In the case of copper applied as a result of the galvanic bath, the difference in the size of the crystals for different variants is also clearly visible. The structure for the variant with a current density of 1 A/dm^2^ is finer.

## 4. Discussion

After conducting the experiments and analyzing the results, the following summary was conceived for the factors considered in the experiment.

The factor that turned out to be the most statistically significant is the current density in the galvanic bath, meaning that this is the factor that had the greatest impact on the robustness of printed circuits. The samples with the variant 2 A/dm^2^ had an SNR that was 6.88 dB higher than that for the second variant. The use of a higher current density in the galvanic bath turned out to be more advantageous for the quality of metallized connections. Although fine-grained crystals are more resistant to mechanical stress, they are characterized by higher resistance. This second parameter turned out to be crucial in terms of the IST.

The number of conductive layers also plays a significant role. After final decisions related to the construction of the test boards, a different type of connection was assigned to each variant of the parameter. The specificity of the PTH’s and microvia’s constructions determined the role that the number of layers plays in the reliability of printed circuits. The plated through-holes proved to be much more reliable than the microvias. The SNR coefficient for samples with PTHs was 6.29 dB higher than that for the second variant of connections. In future studies, an additional advantage would be to also conduct tests on buried holes.

The thickness of the laminate layer has a small effect on the quality of the product. Unexpectedly, it turned out that the thicker the laminate, the better. However, the difference was so small that it was almost imperceptible; the difference in SNR for two variants is equal to 0.41 dB. This may be due to the flexibility of the boards. For both thicknesses, 50 μm and 100 μm, the printed circuits were flexible, even though in the worst case, the entire board was 405 μm thick. Samples made of both polyimide and glass–epoxy laminates were not rigid. In the case of the ISTs, where the work of the material and its resistance to thermal expansion were important, the flexibility turned out to be a huge advantage. Despite the fact that, in the case of a thin laminate, the effect of the volume increasing with increasing temperature was smaller than that in the case of a thick material, the thicker laminate turned out to be slightly better. On the other hand, by referring once again to the previously discussed parameter, it should be noted that two-layer boards performed much better than four-layer ones. Their performance could have been influenced not only by the construction of the connection itself, but also by the thickness of the entire board.

The diameter of the holes also has a small effect on the final reliability. In accordance with the predictions, the larger holes had better results. However, it should be noted here that the obtained experimental results in many cases exceeded 1000 cycles. This is considered very high performance. In order to increase the failure rate of the test boards. the ratio of the hole diameter to its depth, that is, the aspect ratio, would have had to be reduced. Currently, this factor is from 0.125:1 to 0.3:1. In future studies, the aspect ratio should be reduced, for example, by using smaller-diameter holes. In the case of the samples analyzed during this research, the smallest metallized hole had a diameter of 0.3 mm. However, this value is close to the technology limit. Not many manufacturers are able to fabricate metallize holes with a diameter smaller than 0.1 mm. Not only is the drilling process problematic, but so is the metallization of the holes. For this reason, such small holes are difficult to implement. The advantage here is the small thickness of the laminate, which can facilitate the production process itself. Another way to reduce the shape factor is to increase the thickness of the base material. However, it should be noted that this can lead to the basic requirement for the board, its flexibility, not being met. Additionally, in the case of the microvias, their limitation is the maximum value of the aspect ratio, which was 1:1 in this study.

The type of the laminate used led to a similar difference in SNR coefficients to that in the diameter of the holes, about 3 dB, but according to the analysis of variance, this factor is insignificant. Polyimide, as a base material, performed better than the classic glass–epoxy. The small difference may have resulted from the previously discussed properties of the tested circuits, namely their flexibility.

The three factors mentioned above, the laminate thickness, the diameter of the metallized holes and the type of the laminate, did not play a major role in the conducted experiment. The tested printed circuits, with all structure variants, were relatively thin and retained their flexibility. These two properties of the test boards turned out to be of great importance for their reliability during the IST analysis.

In order to confirm the obtained results, an additional verification experiment was conducted. The test samples were manufactured based on the optimized factors. The printed circuit boards for the verification of the results were manufactured with two conductive layers placed on a polyimide laminate, where the laminate layer was 100 μm thick, the diameter of the holes was equal to 0.4 mm and a current density of 2 A/dm^2^ was applied in the galvanic bath. The IST tests were performed for three samples. The obtained results are presented in Table 7 below.

Based on the observation of the results obtained from the verification experiment and the previous ones, it can be concluded that the specified optimal production factors of flexible printed circuits have a positive and noticeable effect on the durability of the electronic connections. Ultimately, the goal of improving the quality of the product is to find the best values of the factors under control that occur in the production process in order to maximize the SRN coefficient. The signal-to-noise ratio for the verification experiment is the best among all those obtained in the conducted experiments. This research showed that the theory behind the design of experiments using the Taguchi method gives the desired effect, i.e., it indicates the optimal solution.

Two parameters turned out to have a major impact on the final result. More research is needed to verify whether the observed trends in the impact of the factors on the reliability will be maintained. For this reason, it is necessary to prepare tests and analyze the factor that has the greatest influence on the results, at three levels; in this study, this factor was the current density in the galvanic bath. In continued research on this topic, it is also worth investigating how holes with smaller aspect ratios will behave, using several variants. An experiment including three or more levels of a factor such as that above will provide a more complete picture of its effect on the robustness of flexible printed circuits.

## 5. Conclusions

Based on the evaluation of the results obtained from the experiment planned using the Taguchi method to examine the reliability of flexible printed circuits using the Interconnect Stress Test, the best variant of the board’s construction was qualitatively determined, which ensured a high-quality final product with a low failure rate. The analysis was performed using the signal-to-noise ratio and an analysis of variance for each factor taken into account during the experiment. The factors that turned out to be the most important in terms of reliability are the number of conductive layers and the current density in the galvanic bath, rather than the type and thickness of the laminate, and the diameter of the metallized holes, which have a relatively small impact on the result. A higher value of current density in the plating bath will result in greater robustness. This research shows that the two-layer structure is more reliable than the four-layer structure, and that plated through-holes are more reliable than the microvias. The thickness of the laminate, the diameter of the vias and the type of base material did not have a significant impact on the test results, because the printed circuit boards in all considered cases were relatively thin and retained their flexibility. In the end, the optimal board construction variant was the use of two conductive layers, on the polyimide laminate, where the laminate layer was 100 μm thick, the hole diameter was equal to 0.4 mm and there was a current density of 2 A/dm^2^ in the galvanic bath.

Based on the optimal factors, the samples for the verification test were produced. Based on the observations of the results obtained from the verification experiment and the previous ones, it can be concluded that the specified parameters of the production process of flexible printed circuits have a positive and noticeable impact on the durability of the connections. They therefore have an innovative impact on the quality of products, and high reliability opens up opportunities for their application in Internet of Things, the space sector, medical industries and Industry 4.0. Moreover, this research methodology shown the high durability of the particular PCB construction variants against thermal stress. The PTHs work well as heat sinks, which is important in the application of GaN technology in high-power semiconductor devices [12]. This experiment achieved the set goal and presented a number of factors that determine the quality of PCBs. The answer to the question of how to achieve high-quality and reliable electronic products while limiting the consumption of raw materials and reducing the amount of waste was obtained.

## Figures and Tables

**Figure 1 sensors-25-00140-f001:**
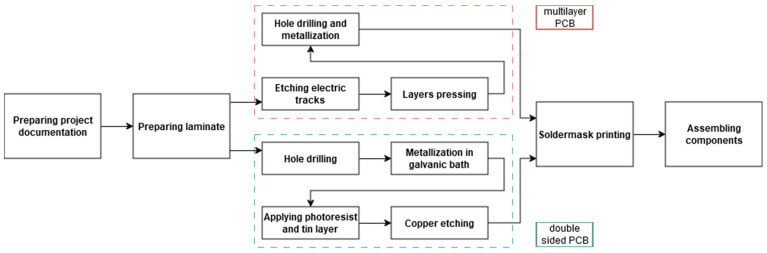
Scheme of production path of PCB.

**Figure 2 sensors-25-00140-f002:**
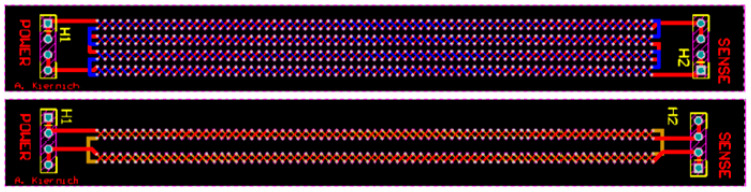
Design of samples, with the two-layer PCB above and the four-layer PCB below.

**Figure 3 sensors-25-00140-f003:**
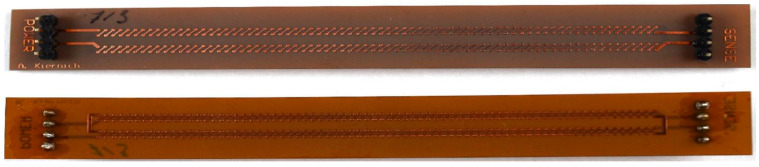
Example of a PCB used as the sample.

**Figure 4 sensors-25-00140-f004:**
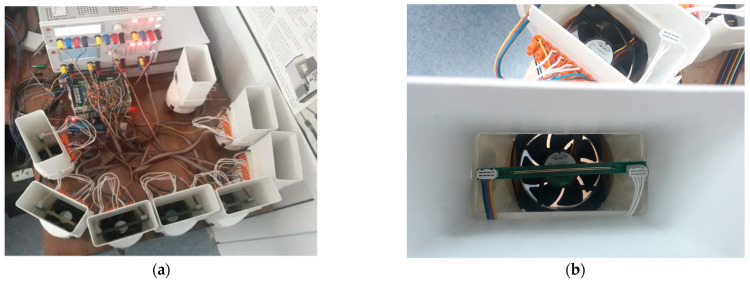
Work station used during the experiment. (**a**) The whole work station; (**b**) a zoomed image of one testing unit.

**Figure 5 sensors-25-00140-f005:**
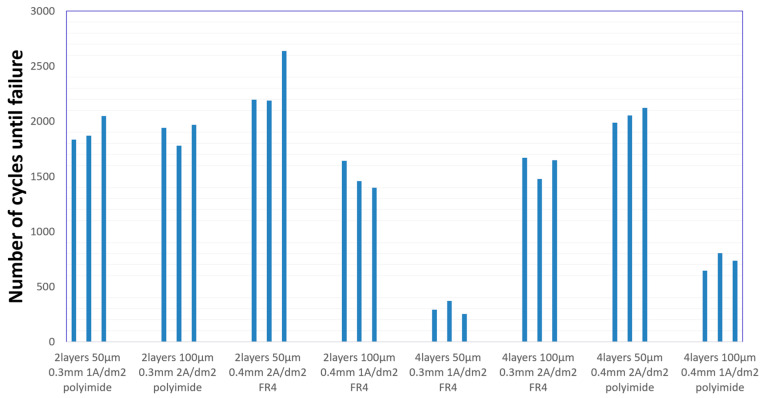
IST results.

**Figure 6 sensors-25-00140-f006:**
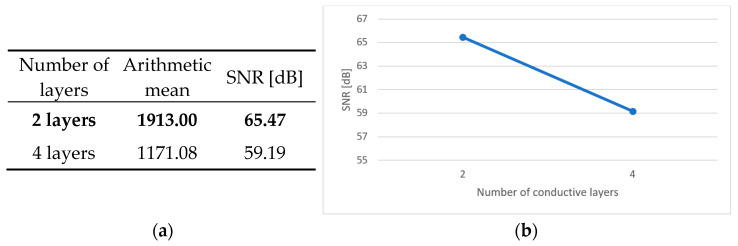
Main effect—number of conductive layers. (**a**) Table showing the following statistics: the arithmetical mean and SNR; (**b**) graph showing the trajectory of the SNR coefficient.

**Figure 7 sensors-25-00140-f007:**
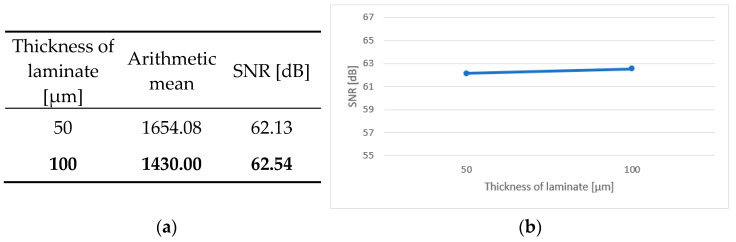
Main effect—thickness of laminate. (**a**) Table showing the following statistics: the arithmetical mean and SNR; (**b**) graph showing the trajectory of the SNR coefficient.

**Figure 8 sensors-25-00140-f008:**
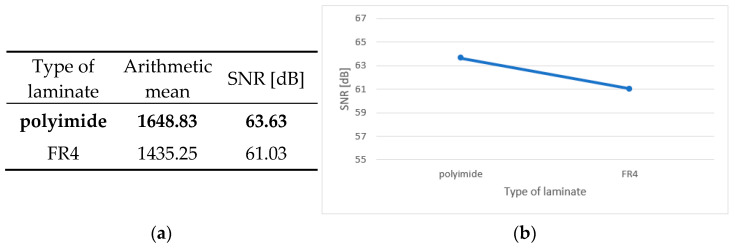
Main effect—type of laminate. (**a**) Table showing the following statistics: arithmetical mean and SNR; (**b**) graph showing the trajectory of the SNR coefficient.

**Figure 9 sensors-25-00140-f009:**
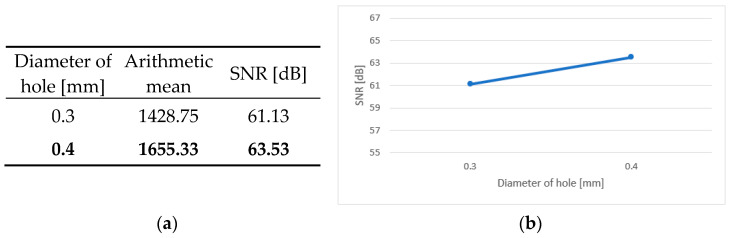
Main effect—diameter of holes. (**a**) Table showing the following statistics: the arithmetical mean and SNR; (**b**) graph showing the trajectory of the SNR coefficient.

**Figure 10 sensors-25-00140-f010:**
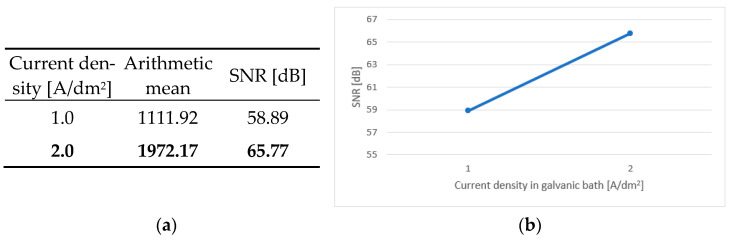
Main effect—current density in the galvanic bath. (**a**) Table showing the following statistics: the arithmetical mean and SNR; (**b**) graph showing the trajectory of the SNR coefficient.

**Figure 11 sensors-25-00140-f011:**
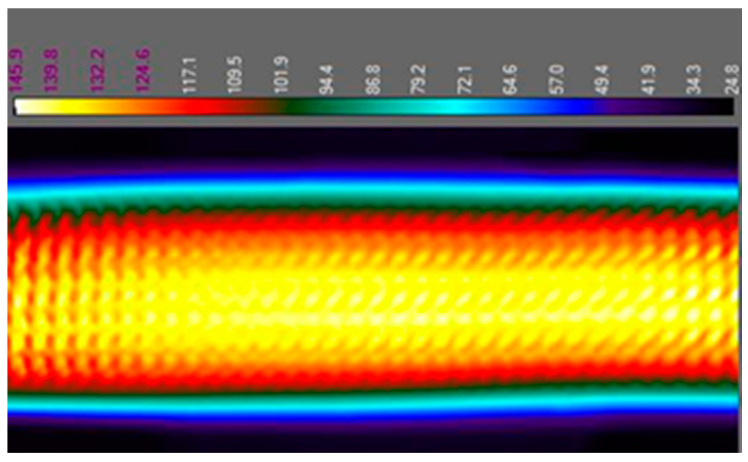
The test board during the heating cycle.

**Figure 12 sensors-25-00140-f012:**
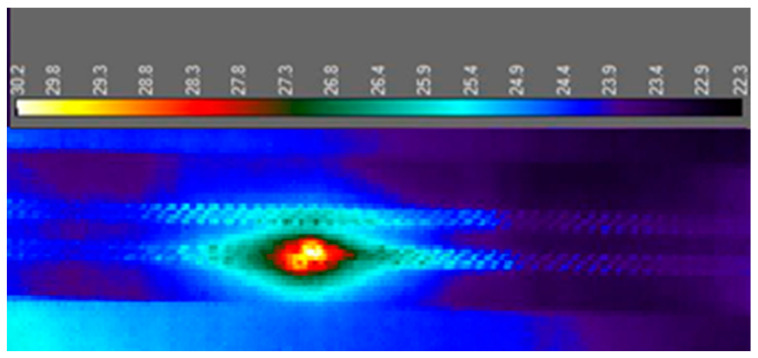
Location of the failure on the test board.

**Figure 13 sensors-25-00140-f013:**
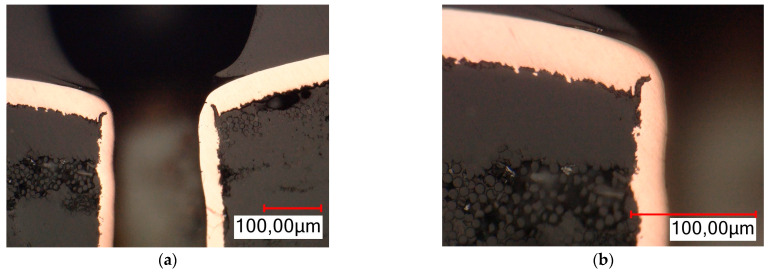
Crack of metallization in PTH (**a**) metallographic cross-section—magnification ×700; (**b**) metallographic cross-section—magnification ×1500.

**Figure 14 sensors-25-00140-f014:**
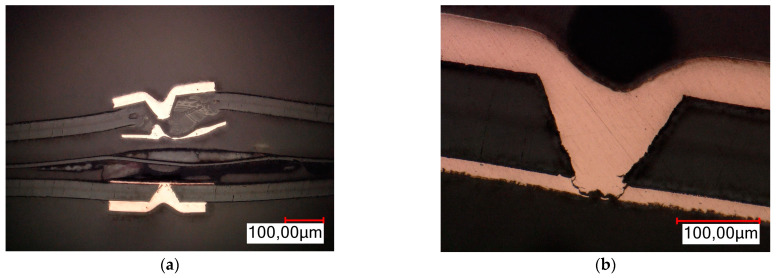
Crack of metallization in microvia (**a**) metallographic cross-section—magnification ×500; (**b**) metallographic cross-section—magnification ×1000.

**Figure 15 sensors-25-00140-f015:**
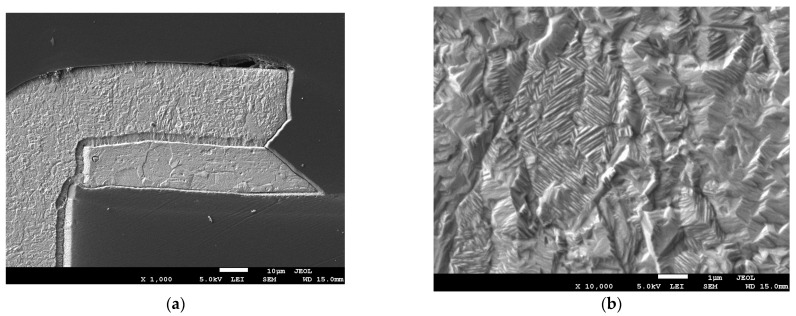
Copper layers with metallization 1 A/dm^2^: (**a**) magnification ×1000; (**b**) magnification ×10,000.

**Figure 16 sensors-25-00140-f016:**
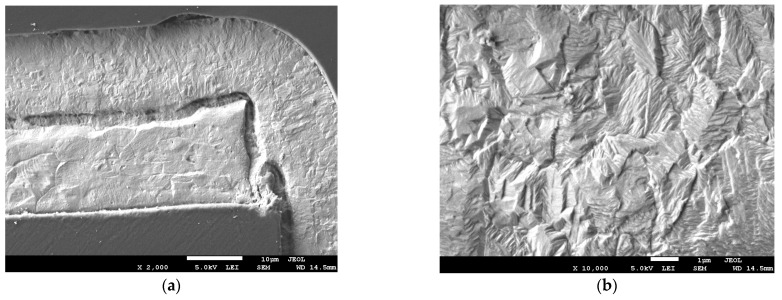
Copper layers with metallization 2 A/dm^2^: (**a**) magnification ×2000; (**b**) magnification ×10,000.

**Table 1 sensors-25-00140-t001:** Factors taken into account during the experiment.

Factor	Levels	
	Level I	Level II
Type of the laminate	polyimide	glass–epoxy
Number of conductive layers	2	4
Thickness of the laminate	50 µm	100 µm
Diameter of plated hole	0.3 mm	0.4 mm
Current density in galvanic bath	1 A/dm^2^	2 A/dm^2^

**Table 2 sensors-25-00140-t002:** Orthogonal array of designed experiment, where A—number of conductive layers; B—diameter of the metallized hole; C—type of laminate; D—laminate thickness; E—variant of the galvanic metallization process of holes.

Number of Experiment					
	A	B	C	D	E
1	2 layers	0.3 mm	polyimide	50 µm	1 A/dm^2^
2	2 layers	0.3 mm	polyimide	100 µm	2 A/dm^2^
3	2 layers	0.4 mm	FR4	50 µm	2 A/dm^2^
4	2 layers	0.4 mm	FR4	100 µm	1 A/dm^2^
5	4 layers	0.3 mm	FR4	50 µm	1 A/dm^2^
6	4 layers	0.3 mm	FR4	100 µm	2 A/dm^2^
7	4 layers	0.4 mm	polyimide	50 µm	2 A/dm^2^
8	4 layers	0.4 mm	polyimide	100 µm	1 A/dm^2^

**Table 3 sensors-25-00140-t003:** Parameters of test samples.

PCB Parameter	Two-Layer PCB	Four-Layer PCB
Dimension	14 mm × 120 mm	14 mm × 120 mm
Thickness	120 µm or 170 µm (depending on variant)	305 µm or 405 µm (depending on variant)
Number of conductive layers	2	4
Number of plated holes	636	636
Total length of the traces	1299 mm	1331 mm
Interconnections	Top–Bottom	Top—1^st^ signal layer 2^nd^ signal layer—Bottom
Diameter of plated holes	0.3 mm or 0.4 mm (depending on variant)	0.3 mm or 0.4 mm (depending on variant)
Thickness of metallization in holes	20 µm	20 µm
Width of tracks	0.254 mm	0.254 mm
Thickness of tracks	35 µm	35 µm

**Table 4 sensors-25-00140-t004:** IPC-TM-650 IST “Method A” review [21].

Test Parameter	Value
Temperature	150
Time of heating	3 min
Time at the maximum temperature	At least 1 s
Failure threshold	Change in resistance by 10%
Cooling method	By forced air
Resistance observation	Continuous
Temperature of sample	Calculation based on measured resistance

**Table 5 sensors-25-00140-t005:** Experiment statistics.

Number of Experiment	Arithmetic Mean	Standard Deviation	SNR [dB]
1	1917.33	93.43	65.62
2	1895.67	83.83	65.53
3	2340.67	210.27	67.29
4	1498.33	104.01	63.45
5	304.00	48.93	49.34
6	1598.00	86.73	64.03
7	2054.33	55.11	66.24
8	728.00	65.46	57.13
**Mean**	**1542.04**	**Sum**	**498.65**

**Table 6 sensors-25-00140-t006:** ANOVA table.

Source of Variation	Sum of Squares	Degrees of Freedom	Mean Square	F Value	P [%]
Number of layers	3,302,642.04	1	3,302,642.04	34.20	31.87
Thickness of laminate	301,280.04	1	301,280.04	3.12	2.91
Diameter of plated hole	308,040.04	1	308,040.04	3.19	2.97
Current density	4,440,180.38	1	4,440,180.38	45.99	42.84
Type of laminate	273,707.04	1	273,707.04	2.83	2.64
Error	1,738,007.42	18	96,555.97		16.77
Total	10,363,856.96	23			100

**Table 7 sensors-25-00140-t007:** Results of the verification test.

	Sample 1	Sample 2	Sample 3	SNR
**Verification test**	2816	2586	2675	68.59

## Data Availability

The raw data of the experiments can be requested from the authors.

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
