# Peer review of "Impact of Particular Stages of the Manufacturing Process on the Reliability of Flexible Printed Circuits"

_sensors, 2024, doi:10.3390/s25010140_

Round 1
Reviewer 1 Report
Comments and Suggestions for Authors The flexible printed circuit board is a very matured manufacturing process. It was used widely and proven to be reliable. From an academic point of view, there is little merit/no need in studying the reliability of this manufacturing process in 2024. Unless there is advancement in flexible printed circuit boards compared to existing ones. I do not see this advancement in this paper. However, despite the lack of importance of this paper. The methodologies used here in the paper are well organized and exemplary. From this point of view, I think this paper is still worth being published. To strengthen the paper, the author needs to address why there is a need to address the reliability of the flexible printed circuit boards in 2024. Flexible printed circuit boards are very reliable in consumables such as an Apple Watch. Generally speaking I have never heard of an apple watch being broken due to a malfunction of its flexible printed circuit board. Why do we need to study the reliability of flexible printed circuit boards?Is the flexible printed circuit board the author studied being used in a very specific market? For example, a printed circuit board heater?
1) Line 73: sever-> severe
2) Figure 2: The 3D structure of the four-layer PCB is not clear to the reader. 3) Table 3: Are the length of the traces the same or different between the two layer PCB and 4 layer PCB? Please provide the info in table 3. 4) Line 313 "Permorfed aging tests" -> "Performed aging tests".It is best to include a time VS temperature VS resistance graph for the aging tests.
5) Please include a zoomed in image of 1 of the 8 testing unit in Figure 4. 6) Line 519. "The structure for the variant with a current density of 1 A/dm2 is finer." By comparing Figure 15(b) and Figure 16 (b), it is not obvious to me. Could you color the boundaries of the crystals ?
7) Need dimension reference/ ruler in Figure 13 and Figure 14.
Author Response
Comment 1: The flexible printed circuit board is a very matured manufacturing process. It was used widely and proven to be reliable. From an academic point of view, there is little merit/no need in studying the reliability of this manufacturing process in 2024. Unless there is advancement in flexible printed circuit boards compared to existing ones. I do not see this advancement in this paper. However, despite the lack of importance of this paper. The methodologies used here in the paper are well organized and exemplary. From this point of view, I think this paper is still worth being published. To strengthen the paper, the author needs to address why there is a need to address the reliability of the flexible printed circuit boards in 2024. Flexible printed circuit boards are very reliable in consumables such as an Apple Watch. Generally speaking I have never heard of an apple watch being broken due to a malfunction of its flexible printed circuit board. Why do we need to study the reliability of flexible printed circuit boards?
Response 1: Modern electronic products used in industries such as automotive, space and medicine are becoming more and more complex. They consist of many components and technologies that may interact in ways that were not predicted during design. As the density of connections and components increases every day, it also creates new challenges. Reliability testing in real-world conditions helps detect problems that may only appear after some use.
Although PCB and FPC technology have been known and developed for a long time, new technologies and solutions are emerging. Also, new restrictions, such as the historic ban on lead in solder and the upcoming ban on Polytetrafluoroethylene (PTFE) in high-frequency boards, create further needs to update our knowledge about the reliability of the used solutions.
Manufacturers using different solutions and tools may also want to check how each element of the production line affects the reliability of their products. Modernization of the plant may also require reestablishing the optimal operating parameters of the equipment in order to obtain the desired quality of the final product.
As technologies become more advanced, user expectations also increase. Consumers expect products that not only work properly, but are also exceptionally durable, effective and reliable over the long term. Even the most advanced technology must undergo testing to ensure it meets these high requirements. This problem concerns not only private consumers, but also fields of science and technology where reliability is crucial. We create devices that are to be placed in difficult conditions (space rockets, sensors in hard-to-reach and dangerous environments, etc.).
In many applications, such as automotive, aviation, medicine and the defense industry, the reliability and quality of products are crucial to the safety of users. Even the smallest technological defects can lead to serious consequences, which is why these tests are necessary to ensure that the product does not pose a risk to health or life.
Referring to your example with Apple Watch. Everyone will agree that it is a highly technologically advanced device, generally known and respected. But everyone also knows the location of the nearest Apple or other manufacturer's service center because these devices, even though they have been developed for many years, can still break down.
To sum up, I strongly believe that problem of robustness of PCBs is still present-day issue, which is worth considering.
Comment 2: Is the flexible printed circuit board the author studied being used in a very specific market? For example, a printed circuit board heater?
Response 2: Tested PCBs are specially designed boards used only for the purpose of the experiment. In normal manufacturing lines samples are placed on technological formats, which also contain printed circuits that are final product. The entire form goes through the same production line and is subjected to the same processes. Therefore, the results of IST tests performed on samples are reliable and apply to all boards produced in a given plant or on a given production line. In other words flexible printed circuit boards that were studied do not have any special application in the market, they are producted only for the purpose of IST.
Comment 3: Figure 2: The 3D structure of the four-layer PCB is not clear to the reader.
Response 3: Design of the PCBs shown in the Figure 2 are printscreens from the application used to design the samples. I do not have many tools to improve the quality of the image since it is limited by the capabilities of the software.
Comment 4: Table 3: Are the length of the traces the same or different between the two layer PCB and 4 layer PCB? Please provide the info in table 3. and Please include a zoomed in image of 1 of the 8 testing unit in Figure 4. and Need dimension reference/ ruler in Figure 13 and Figure 14.
Response 4: All recommended additions were included in the paper.
Comment 5: It is best to include a time VS temperature VS resistance graph for the aging tests.
Response 5: During tests all the information regarding temperature and resistance change in time were gathered by software. However research conclusions and additional analyzes (anova, metallographic cross-section) were based on the number of cycles that the samples survived until the failure. Changes in resistance over the course of exposures were considered insignificant, so they were not included in the results and in the paper.
In the attachment you may find the paper with corrected minor errors in English highlitghted in yellow.

Reviewer 2 Report
Comments and Suggestions for Authors
This is an interesting study that can help designers determine that the type of PCB that is acceptable for their application.
I would like to know more about the manufacturing of the boards. How and where were they made - on a large production line or a small pilot line or a lab setting?
I would also like to see some discussion of how transferable these results are to other manufacturing lines. For example, the current density in the galvanic bath was found to me the factor that had the most statistical significance. Would you expect that this is true across all manufacturing line? Does it depend on the type of equipment used for the galvanic bath?
I would also like to see some discussion and references relative to the findings about plated through holes and microvias. Is this a common finding that plated through holes are more reliable than microvias?
I believe each set of parameters was run once. Is there information on how much process variation there is in the process overall? In other words if the same conditions were run multiple times - how much variation would there be in the results.
Comments on the Quality of English LanguageSome minor grammatical (English) issues and misspellings. For example. Line 313 - title of 2.4, I believe should read "Performed aging tests"
Another example in line 553 "the larger the hole coped better." I'm not clear on the meaning of this phrase.
I think it would be helpful to have a very close and careful editing of the paper for English. Some of the phrasing is a bit awkward - yet it is easily understood.
Author Response
Thank you very much for your comments and opinion. We especially appreciate the positive feedback on the paper. We would also like to refer to five listed issues.
Comment 1: I would like to know more about the manufacturing of the boards. How and where were they made - on a large production line or a small pilot line or a lab setting?
Response 1: The samples were manufactured in the Łukasiewicz Research Network – Tele and Radio Research Institute. The Institute has small production line with specialization of small series. The production department in the Institute is capable to produce different types of PCBs, multilayered on different laminates. The Institute has many decades of experience in production of PCBs and achvements in this field.
Comment 2: I would also like to see some discussion of how transferable these results are to other manufacturing lines. For example, the current density in the galvanic bath was found to me the factor that had the most statistical significance. Would you expect that this is true across all manufacturing line? Does it depend on the type of equipment used for the galvanic bath?
Response 2: All manufacturers have to meet standards and requirements. For example, all factories and their products must meet the worldwide recognized and applied IPC standard. Meeting these restrictions ensures that, regardless of the manufacturer, each product will meet the minimum requirements and we will receive a product with the same functionality. On the other hand true statement is that each production line is different. In case of galvanic bath, different producers can use different bath chemistry. If we expect excactly the same results, we must have the same equipment. However there are some general physical rules independent from type of production line. Regardless of the bath used, a higher current density will be faster process and will result in larger crystals. Since the authors see the size of the copper crystals as the main reason for the higher reliability at applied 2 A/dm2 in galvanic bath, we expect that the use of higher current density in other factories will also result in greater durability of the final product.
Comment 3: I would also like to see some discussion and references relative to the findings about plated through holes and microvias. Is this a common finding that plated through holes are more reliable than microvias?
Response 3: The plated through holes are larger in size compared to microvias. A larger diameter of PTH can result in better heat dissipation, which reduces local thermal stresses. In the case of microvias, a small diameter that narrows at the neck can lead to greater stress concentration, which increases the risk of the failure. The crosssection analysis clearly shows that the failure of the microvia occured near the neck. PTHs are generally larger and allow for thicker layers of copper in the hole. A thicker layer can improve resistance to temperature changes because the material can absorb and dissipate heat better. Plated through holes are also relatively simple to manufacture, which means they are less susceptible to technological errors that can affect their robustness. In the case of microvias, the quality of the technological process is essential, because small errors can lead to poor quality of the connection. On the other hand, in HDI (High-Density Interconnect) PCBs, microvias have become a necessity, so it is extremely important to ensure good quality of the connection.
According to this article https://doi.org/10.1108/03056120911002361 "Single and 2 stack microvias are generally the most robust type of copper interconnection used in HDI applications, 3 stack and 4 stack require greater discipline to assure product reliability. Ranking the inherent reliability of 3 stack and 4 stack structures to other interconnects like plated through holes, blind, or buried vias, may need to be reconsidered in future reliability test programs."
Comment 4: I believe each set of parameters was run once. Is there information on how much process variation there is in the process overall? In other words if the same conditions were run multiple times - how much variation would there be in the results.
Response 4: Production of PCB is very complex process where hundreds of factors and operations are involved. Laminate during production goes through such processes as hole drilling, galvanic bath, etching copper, pressure welding layers and many, many more. Despite the complexity of the process and a wide range of variations during PCB production, only those factors that were established during design of experiment were changed. If the samples were produced on the same production line while maintaining the same values of all factors, we can assume that the results would be the same.
In the attachment you may find the paper with corrected minor errors in English highlitghted in yellow. Figure 4(b) was also added.

Reviewer 3 Report
Comments and Suggestions for Authors
The manuscript employs the Taguchi method to design experiments aimed at investigating the impact of various factors on the reliability of flexible printed circuit boards (FPCs) during production. Impressively, the authors analyze five production factors—laminate type, number of conductive layers, laminate thickness, plated hole diameter, and current density of the electroplating bath—and evaluate their reliability under different conditions using the interconnect stress test (IST) method. Overall, this study provides valuable insights into improving the quality and reliability of FPCs for IoT terminals and Industry 4.0 applications. The manuscript is suitable for publication after addressing the following minor revisions:
- Regarding the selection of laminate types, apart from polyimide and glass epoxy resin, were other materials considered? Could other materials potentially impact the reliability of FPCs?
- In the experimental design, a fractional factorial design (L8 array) was used. Have the authors considered increasing the sample size and factor levels to enhance the statistical significance of the results?
- From a practical production perspective, could there be other potential factors, such as production line speed, that might significantly influence the reliability of FPCs?
- Figures 11 and 12 appear slightly blurry. It is recommended to replace them with clearer images to facilitate better understanding by the readers.
- Regarding error analysis, have the authors discussed potential sources of error during the experimental process, such as material batch variations or minor fluctuations in electroplating equipment?
Author Response
Thank you very much for your comments and opinion. We especially appreciate the positive feedback on the paper. We would also like to refer to five listed issues.
Comment 1: Regarding the selection of laminate types, apart from polyimide and glass epoxy resin, were other materials considered? Could other materials potentially impact the reliability of FPCs?
Response 1: Glass-epoxy laminate is the most popular laminate nowadays. Especially FR4 variant. We can meet this laminate in most PCBs. On the other hand polyimide laminate is very popular choice for flexible printed circuit boards. Since the research was limited by time and cost it was decided to selected only this two highly successful materials in the market. Of course there are plenty other laminates that can be used in the production of PCBs and have impact on the reliability of the product. For example another material used in FPC is widly known also outside the electronics industry, polyester. Another interesting material to test would be laminate consisting aluminium (it could be an extremely durable material during thermal cycle tests) or laminate with Polytetrafluoroethylene, PTFE, material used in high-frequency applications.
Comment 2: In the experimental design, a fractional factorial design (L8 array) was used. Have the authors considered increasing the sample size and factor levels to enhance the statistical significance of the results?
Response 2: The conducted reserch was part of my master thesis . For this reason the experiment was limited by time and costs. After discussion within the team it was decided that five factros two levels each will be sufficient to obtain interesting results and have small insight into problem of robustness of the PCB. It was also decided that testing three samples of each variant would be statistically significant and would allow for the exclusion of random error. However in future research number of factor levels should and will be increased in order to ensure that the observed trends remain valid.
Comment 3: From a practical production perspective, could there be other potential factors, such as production line speed, that might significantly influence the reliability of FPCs?
Response 3: Production of PCB is very complex process where hundreds of factors and operations are involved. Laminate during production goes through such processes as hole drilling, galvanic bath, etching copper, pressure welding layers and many, many more. Let's take hole drilling as an example. Holes can be drilled with different speed, they can be drilled by different tools, by mechanical blade or by laser. These factors are the first that came to my mind. During planning the experiment it was decided to choose only five factors that in our opinion have the biggest influence on the reliability of PCBs and on which designer can have direct impact or indirect by consulting the manufacturer. In the next experiment developing the issue of robustness of PCBs, another factor taken into account may be the thickness of copper in the tracks and plated holes. We think it can important in durability of samples during IST. As it was mentionted earlier, it was decided to investigate only five factors because of the limit in time and costs. Thats why only five factors were taken into account, after strict selection.
Comment 4: Figures 11 and 12 appear slightly blurry. It is recommended to replace them with clearer images to facilitate better understanding by the readers.
Response 4: Figures numbered as 11 and 12 are printscreens fromt he computer connected directly to the thermal camera. Unfortunetelly it is impossible to improve the quality of the image since it is limited by the capabilities of the camera and the application.
Comment 5: Regarding error analysis, have the authors discussed potential sources of error during the experimental process, such as material batch variations or minor fluctuations in electroplating equipment?
Response 5: All manufacturers have to meet standards and requirements. For example, all factories and their products must meet the worldwide recognized and applied IPC standard. Meeting these restrictions ensures that, regardless of the manufacturer, each product will meet the minimum requirements and we will receive a product with the same functionality. The authors have not discussed sources of errors during the experiment process since it was assumed that the fluctuations are within the boundaries accepted by the standards. The samples were produced in the same production line, in the same time. Because of that it can be assumed that all samples were subjected to the same processes and possible fluctuations were negligible.
In the attachment you may find the paper with corrected minor errors in English highlitghted in yellow. Figure 4(b) was also added.
